# Synaptic Plasticity Is Predicted by Spatiotemporal Firing Rate Patterns and Robust to In Vivo-like Variability

**DOI:** 10.3390/biom12101402

**Published:** 2022-10-01

**Authors:** Daniel B. Dorman, Kim T. Blackwell

**Affiliations:** 1Interdisciplinary Program in Neuroscience, George Mason University, Fairfax, VA 22030, USA; 2Department of Bioengineering, Volgenau School of Engineering, George Mason University, Fairfax, VA 22030, USA

**Keywords:** calcium, LTP, synaptic plasticity, striatum, learning

## Abstract

Synaptic plasticity, the experience-induced change in connections between neurons, underlies learning and memory in the brain. Most of our understanding of synaptic plasticity derives from in vitro experiments with precisely repeated stimulus patterns; however, neurons exhibit significant variability in vivo during repeated experiences. Further, the spatial pattern of synaptic inputs to the dendritic tree influences synaptic plasticity, yet is not considered in most synaptic plasticity rules. Here, we investigate how spatiotemporal synaptic input patterns produce plasticity with in vivo-like conditions using a data-driven computational model with a plasticity rule based on calcium dynamics. Using in vivo spike train recordings as inputs to different size clusters of spines, we show that plasticity is strongly robust to trial-to-trial variability of spike timing. In addition, we derive general synaptic plasticity rules describing how spatiotemporal patterns of synaptic inputs control the magnitude and direction of plasticity. Synapses that strongly potentiated have greater firing rates and calcium concentration later in the trial, whereas strongly depressing synapses have hiring firing rates early in the trial. The neighboring synaptic activity influences the direction and magnitude of synaptic plasticity, with small clusters of spines producing the greatest increase in synaptic strength. Together, our results reveal that calcium dynamics can unify diverse plasticity rules and reveal how spatiotemporal firing rate patterns control synaptic plasticity.

## 1. Introduction

Synaptic plasticity—the activity-dependent modification of synaptic strength—is widely hypothesized as the neural substrate of learning and memory throughout the brain [1]. For instance, synaptic plasticity in mammalian striatum [2], cortex [3], and hippocampus [4] have been linked to procedural, sensorimotor, and associative learning and memory, respectively. Learning requires that repeated experiences produce a stable, persistent change in synaptic connections which in turn produce stable neural activity and behavioral responses [5,6]. In vivo experiments have revealed changes in synaptic strength and size of dendritic spines (sites of synaptic input) [7,8,9,10]. However, evidence for stable synaptic changes primarily comes from in vitro brain-slice experiments using precisely repeated stimulation patterns, which reveal that stimulus timing, frequency, and synaptic location controls development of long term potentiation (LTP) or long term depression (LTD) [11,12,13,14,15]. Yet, it is unclear if in vitro plasticity discoveries that used precise stimulation patterns are reproducible in the highly variable neural activity conditions in vivo during natural learning and behavior. Indeed, one of the great unsolved questions in neuroscience is whether long-lasting synaptic plasticity occurs in vivo given variable neural activity.

Cortical neurons exhibit significant trial-to-trial variability in vivo in response to the same external sensory stimulus [16,17]. Trial-to-trial variability includes variance both in the timing of spikes and in firing rate. Thus, a post-synaptic neuron experiences variability at each of its thousands of synaptic inputs, which produces highly variable output spiking. In the in vitro case, not only are pre-synaptic and post-synaptic spike timing much less variable during repetition, but also the spatial pattern on the dendritic tree of the repeatedly activated synapses is likely less variable in vitro as well. Though the issue of plasticity with spike time variability has been addressed [18,19], the role of spatial patterns was ignored.

Experiments suggest that spatial organization of synaptic inputs on the dendritic tree are important for plasticity. For instance, in vitro, cortical synaptic plasticity has been shown to depend not only on rate and timing, but also on cooperativity of inputs [11], which is influenced by the spatial organization of synaptic inputs on the dendritic tree [20,21,22,23,24]. Yet, the effect of variability on spatial cooperativity under in vivo-like conditions is unclear.

In spiny projection neurons (SPNs) of the striatum, the input nucleus of the basal ganglia and the focus of this paper, dopamine and calcium activated signaling pathways interact to produce synaptic plasticity. In vitro, a dopamine signal occurring within 1–2 s of synaptic activation produces long term potentiation [25]. In vivo, a pause in acetylcholinergic neuron activity coordinated with phasic dopamine activity and depolarization produces LTP when given subsequent to SPN upstates, whereas LTD occurrs when either dopamine or depolarization alone follow SPN upstates [26]. In all cases, the cortical inputs producing the upstate in vivo or synaptic activation in vitro activate calcium signaling pathways that lead to synaptic plasticity. In other words, the calcium signaling may provide an eligibility trace for corticostriatal plasticity (either LTP or LTD) underlying reinforcement learning for goal-directed or habitual behavior [27,28].

One approach to investigate how in vivo variability influences synaptic plasticity is to evaluate calcium signals in response to in vivo spike trains. The spatiotemporal dynamics of the calcium signal, and not just peak calcium, determine the occurrence and direction (potentiation or depression) of plasticity [29,30,31]. Specifically, both models and experiments suggest that a prolonged, but moderate elevation produces LTD by activation of protein phosphatases or production of endocannabinoids, and a shorter but higher amplitude calcium produces LTP through activation of protein kinases. Several models are able to account for a diversity of synaptic plasticity data [18,32,33,34,35]; however, they utilize phenomenological descriptions of calcium dynamics or downstream signaling pathways. We previously showed that a biophysical model of calcium dynamics in an SPN, which has been carefully adjusted to match experimental data, could predict LTP and LTD outcomes of several in vitro plasticity experiments using a plasticity rule with two amplitude and two duration thresholds [36]. According to this rule, if the calcium exceeds the LTP threshold for the LTP duration criterion, the synaptic strength increases. If the calcium remains between the LTD and LTP thresholds for the LTD duration criterion, then synaptic strength decreases. Here, we extend that work and evaluate whether our calcium-based plasticity rule in a neuron with dendritic morphology predicts whether plasticity is robust to spatiotemporal trial-to-trial variability and which specific spatiotemporal patterns produce LTP or LTD. We find that persistent plasticity is robust to trial-to-trial variability, and we also demonstrate that small clusters of synaptic inputs are more likely to produce LTP.

## 2. Methods

We developed a biologically-constrained computational SPN model with multiple ion channels identified in SPNs, real dendritic morphology, explicit dendritic spines, sophisticated calcium dynamics, and a calcium-based plasticity rule. Model parameters were determined using our parameter optimization software to fit the model to electrophysiological current injection data [37]. Synaptic parameters and calcium parameters were constrained by experimental responses to synaptic input [38,39,40]. To generate in vivo-like synaptic inputs, we obtained anterior lateral motor cortical spike trains from the CRCNS.org (accessed on 23 March 2019) repository. The anterior lateral motor cortex was selected because those pyramidal neurons project to the striatum, and the spike trains were collected during a learning task known to involve the striatum [41,42]. The model was simulated with these in vivo spike trains to investigate whether plasticity occurs with in vivo-like activity.

### 2.1. Spn Model Morphology and Passive Membrane Properties

A validated, biophysical SPN model we previously published [43] was updated for the MOOSE simulator (https://moose.ncbs.res.in, accessed on 6 June 2022 (version 3.2.0.dev20190806)). We used a D1 SPN morphology obtained from the Luebke repository [44] on neuromorpho.org, accessed on 6 June 2022 (ID: Apr29IR2b) [45]. Dendritic spines were modeled both implicitly and explicitly. Explicit dendritic spines were created for synaptic inputs and calcium dynamics at a density of 0.1 spines/µm on dendritic branches greater than 25 µm from the soma (for simulations evaluating spike train variability) or between 0.75 and 2.75 spines/µm on a subset of dendritic segments greater than 25 µm from the soma (for simulations using clustered synaptic input). Each spine had a cylindrical head (0.5 µm diameter, 0.5 µm length) and neck (0.12 µm diameter, 0.5 µm length). The density of explicitly modeled spines is not representative of the full spine density observed anatomically, and modeling the full spine density would be computationally intensive. Therefore, the implicit effect of dendritic spines (those not explicitly modeled) on passive membrane properties and dendritic channel densities was modeled by compensating dendritic membrane resistivity (RM), membrane capacitivity (CM), and axial resistivity (RA), as well as channel maximal conductance values [46] using a distance dependent function fit to experimentally observed spine density versus distance from the soma [47]. Values for RM, CM, and RA were set to 6.02 $\Omega m^{2}$, 0.011 F/m${}^{2}$, and 1.3 Ωm, respectively, based on automatic parameter optimization.

### 2.2. Voltage Gated Ion Channels

As previously described [43], the model incorporates the following voltage gated sodium and potassium ion channels that have been observed in SPNs: a fast sodium channel (NaF) [48]; fast (Kaf/Kv4.2) [49] and slow (Kas/Kv1.2) [50] transient potassium channels; an inwardly rectifying potassium channel (Kir2) [51]; a resistant persistent potassium channel (Krp; also called delayed rectifier) [52]; a big conductance voltage- and calcium-activated potassium channel (BK) [53]; and a small conductance calcium-activated potassium channel (SK) [54]. Six voltage gated calcium channels (VGCCs) are also included: R-type (CaR/Cav2.3) [55,56], N-type (CaN/Cav2.2) [57,58,59], two L-type (CaL1.2/Cav1.2 and CaL1.3/Cav1.3) [57,58,60], and two T-type (CaT3.2/Cav3.2/α1H and CaT3.3/Cav3.3/α1I) [61]. We newly added a calcium activated chloride channel (CaCC) based on the ANO2/TMEM16B channel, which has been observed in SPNs and allows better fit of the AHP waveform [62,63]. Channel kinetic equations are the same as our previously reported model, but parameters were updated during the parameter optimization by allowing half-activation voltages and time constants to be modified by the parameter optimization algorithm.

Model optimization was done using ajustador software [37,64] that was updated to allow variation of channel kinetics, in addition to channel conductances and membrane properties, in order to fit the model response to experimental data for injected current steps. Briefly, the parameter optimization algorithm utilized a covariance matrix adaptation with evolutionary strategy (cma-es) to vary model parameters [65], and a feature-based fitness metric to compute the fitness between model and experimental data based on features such as number of spikes, action potential height, action potential width, spike timing, steady state membrane voltage, depolarization and hyperpolarization time constants, and afterhyperpolarization waveform. Channel conductances and membrane properties were varied within a linear range, while kinetic parameters (half activation voltage and time constant) were varied with a multiplicative parameter between 0.5 and 2. Optimizations utilized the Neuroscience Gateway Portal [66]. Table A1 provides the conductance values for all channels.

### 2.3. Calcium Dynamics

Intracellular calcium concentration was modeled with diffusion, calcium-binding proteins (buffers), and transmembrane calcium pumps. One-dimensional radial diffusion of buffers and calcium was modeled with concentric shells in dendrites, while spine heads and necks implemented one-dimensional axial diffusion in cylindrical slabs connected from the spine neck to the submembrane shell of the dendritic shaft [67]. Transmembrane calcium extrusion mechanisms—plasma membrane calcium ATPase (PMCA) in every compartment and sodium-calcium exchanger (NCX) limited to spines—were modeled with Michaelis-Menten kinetics. The calcium-binding proteins included calbindin and calmodulin (N and C terminals), which could diffuse between calcium compartments, and an endogenous immobile buffer (representative of several potential biological mechanisms that buffer calcium without diffusing) [68,69]. Pump density and buffer concentration parameters were constrained by calcium imaging data [38,39,40].

The sources of calcium influx in the model included the voltage gated calcium channels and the NMDAR synaptic channel. Calcium concentration in the submembrane shell was used for calcium-dependent channel activation (BK, SK, CaCC) or inactivation (R-, N-, and L-type calcium channels).

### 2.4. Synaptic Channels

Excitatory NMDAR and AMPAR synaptic channels were included on spine heads, and each cortical synaptic input activated both the NMDAR and AMPAR. Inhibitory GABA_A_ channels were included on the dendritic shaft with a density of 4 synapses per dendritic compartment. Activation of these synapses is described in Section 2.6 Synaptic Inputs. All three types of synaptic channels were modeled with dual exponential kinetics, and the NMDA channel included voltage-dependent magnesium blocking. Synaptic conductance parameters were constrained using experimental measures of response to synaptic input [39,70,71].

### 2.5. Plasticity Rule

The calcium-based plasticity rule used dual amplitude and duration thresholds for spine calcium concentration to determine LTP and LTD, as we previously described [36]. For LTD, spine calcium had to exceed the amplitude threshold, $TA_{D}$, of 0.33 μM for longer than the duration threshold of 28 ms, while for LTP the amplitude threshold, $TA_{P}$, was 0.53 μM and the duration threshold was 3.3 ms. Once thresholds were exceeded, the synaptic weights were updated at each time step toward their maximum or minimum. For LTP, while thresholds were exceeded, synaptic weight was increased according to:(1)$$\Delta w\left(t\right)=\min\left\{\gamma_{Pmax},\gamma_{P}\;\cdot \;\left({\left[Ca^{2+}\right]}_{sp}\left(t-1\right)-TA_{P}\right)\right\}\;\cdot \;\sqrt{1-\frac{w\left(t-1\right)-w_{min}}{w_{max}-w_{min}}}$$
where $\gamma_{P}$ is a gain factor for potentiation, $\gamma_{Pmax}$ is the maximum allowable gain value, ${\left[Ca^{2+}\right]}_{sp}$ is spine calcium concentration, $TA_{P}$ is the amplitude threshold for potentiation, $w_{max}$ is maximum allowable synaptic weight (2.0), and $w_{min}$ is minimum allowable synaptic weight (0). Similarly, when LTD thresholds were exceeded, weight was decreased according to:(2)$$\Delta w\left(t\right)=-\min\left\{\gamma_{Dmax},\gamma_{D}\;\cdot \;\left({\left[Ca^{2+}\right]}_{sp}\left(t-1\right)-TA_{D}\right)\right\}\;\cdot \;\sqrt{\frac{w\left(t-1\right)-w_{min}}{w_{max}-w_{min}}}$$
where $\gamma_{D}$ and $\gamma_{Dmax}$ are the gain and maximum allowable gain for depression, and $TA_{D}$ is the amplitude threshold for depression.

### 2.6. Synaptic Inputs

In vivo cortical spike trains were obtained from a CRCNS.org (accessed on 23 March 2019) repository, consisting of recordings from 25 simultaneously recorded anterior lateral motor cortex pyramidal neurons with 90 repeated trials [72]. A single initial trial of model input consisted of 200 spike trains (Figure 1A,B), which were selected from 22 similar behavioral trials (excluding neurons that were inactive within a trial) in order to preserve within-trial correlations between neurons.

To generate controlled trial-to-trial variability of spike timing, the initial trial was repeated 10 times with random jitter of each spike on each repetition. Trial-to-trial variability was limited to standard deviation of spike timing while constraining the same total number of spikes per spike train within each trial. The random jitter was generated from a truncated normal distribution using a standard deviation of 1, 10, 100, or 200 ms (truncated such that no value outside the start time or end time of the trial was selected). Experiments consisted of 10 trials with a single standard deviation. Trials were separated by a 1 s intertrial interval, during which time membrane potential returned to resting potential and spiking activity ceased.

We also implemented a different type of trial-to-trial variability that allows variability of spike rate for individual synapses, but maintains the same spike timing to the neuron as a whole. Variable spike rate was implemented by randomly moving individual spikes from one pre-synaptic input train to another, with the probability of each spike being moved between 10 and 100%. To maintain the distribution of spike counts per trial, the randomly selected target train probability was weighted by the number of spikes of the target trains. This ensured both the overall spike timing pattern to the whole neuron was maintained as well as the distribution of spike counts to synapses, while allowing small variations in spike rate for each synapse from trial to trial.

To evaluate the role of spatial pattern, a single trial was repeated using different mappings of spike trains to clusters of synapses. Each simulation used a different number of spines per cluster and a different cluster length (dendritic distance between the most separated pair of spines in the cluster). Figure 2 shows two different clustering examples. To evaluate the role of correlations between pre-synaptic neurons, we then repeated this set of trials using spike trains in which the interspike intervals had been shuffled. This preserved the overall instantaneous firing rate to the neuron, and isolated the contribution of spatial patterns.

In addition to excitatory inputs, the neuron also received two types of inhibitory inputs. All inhibitory inputs were constructed from Poisson processes with mean firing rates consistent with both striatal fast spiking interneurons (FSIs) and low threshold spiking interneurons (LTSIs). FSI inputs were generated with a mean firing rate of 12 Hz [73] and densely targeted soma and proximal dendrites (within 80 microns of the soma), while LTSI inputs were generated with a mean firing rate of 8 Hz [74] and targeted distal dendrites (greater than 80 microns from the soma). Inhibitory trains were active for the entire 21 s experiment duration of the 10 repeated trials.

### 2.7. Analysis

Analysis of simulations used Python3 and the following python packages: Numpy, Scipy, Pandas, Scikit-learn, Statsmodels, and Matplotlib. For analysis relating spatiotemporal input patterns to magnitude and direction of synaptic weight change, we introduce a “weight-change triggered average,” which is constructed by grouping synapses per trial into bins based on the value of the weight change at the end of a trial, computing the instantaneous firing rate of the spike train input to each given synapse, and averaging the instantaneous firing rates within each bin of synaptic weight change. This is analogous to a spike-triggered average [75], but using the continuous valued, trial-level weight change instead of the spike.

Random forest regression was applied to the instantaneous firing rate to a synapse, instantaneous firing rate to neighboring synapses, cluster length, and the distance from synapse to soma. Random forest regression applies a non-linear method for predicting the weight change from a set of features, such as pre-synaptic firing rate. The prediction is determined from a set of hierarchical rules, where each rule partitions the weight change based on a single feature. A non-linear method was required because of the non-linear relationship between weight change and pre-synaptic firing. The instantaneous firing rate of the input was discretized into 1-5 features (time samples), where 1 time sample calculates the mean firing rate for the entire trial, and 5 time samples calculates mean firing frequency for subsequent 200 ms time intervals. To determine the optimal features for predicting the weight change, a random forest regression was performed for each combination of features. For each set of simulations, the trial-level weight changes were randomly subdivided into a testing set (1/N of the data) and a training set (the remainder of the data); subdividing the data and performing the random forest regression was repeated N times. Then analysis of variance was used to determine which combination of features best predicted the weight change.

### 2.8. Code Accessibility

All code for simulation and analysis is available on github (https://github.com/neurord/moose_nerp/releases/tag/v2.2 (accessed on 6 June 2022) and modelDB (http://modeldb.yale.edu/267552 (accessed on 6 June 2022) Code used for parameter optimization is available on github (https://github.com/neurord/ajustador/releases/tag/v2.1 (accessed on 6 June 2022).

## 3. Results

### 3.1. Data-Driven SPN Model Exhibits Calcium-Based Synaptic Plasticity for In Vivo-like Inputs

To investigate the effects of in vivo-like corticostriatal synaptic inputs on corticostriatal plasticity, we created a realistic biophysical SPN model [76] with a calcium-based plasticity rule [36] and simulated in vivo-like synaptic input patterns. The multi-compartment, multi-ion channel model was optimized to fit electrophysiological data using an extended version of a parameter optimization algorithm we developed [37,64]. The validated model reproduced the characteristic electrophysiological responses of SPNs to both current injection and synaptic inputs in vitro.

Trial-to-trial variability is regularly observed in in vivo spike train recordings but the effect of variability on plasticity is unclear. To simulate synaptic plasticity in response to trial-to-trial variability, we obtained spike train recordings from the anterior lateral motor cortex from a published dataset [72] from which we constructed synaptic inputs to the model. An initial single trial of corticostriatal inputs was constructed from all the spike trains of 22 behaviorally similar experimental trials to generate sufficient synaptic drive while maintaining, as much as possible, potential within-trial correlations between neurons present in the dataset. The initial one-second trial (shown as raster plot and peri-stimulus time histogram in Figure 1A,B) produced depolarization and spiking in the SPN (Figure 1C) with a firing rate consistent with in vivo observations.

The first question addressed is whether a calcium-based synaptic plasticity rule that was derived to explain STDP data is sufficiently general to produce synaptic plasticity in response to spatiotemporally distributed synaptic inputs. We simulated synaptic weight changes in response to in vivo spike trains with trial-to-trial variability. We used our calcium-based plasticity rule that can reproduce results from several spike-timing dependent plasticity experiments on SPNs in vitro [36]. Crucially, this rule is entirely based on data-constrained spine calcium dynamics, not relative spike timings, so it is a general rule that encompasses both frequency-based and spike-timing plasticity rules. The synaptic plasticity rule includes two of the previously recommended mechanisms [77]: temporal properties of calcium signaling and locally spreading (dendritic) signals, while making no assumptions as to the source of the calcium.

Cortical spike trains indeed produced synaptic plasticity with our calcium-based plasticity rule. We examined the spine calcium concentration and the synaptic weight of every synapse in response to a single 1-second trial of randomly distributed in vivo spike trains. We found that, at the end of a single trial, some synapses exhibited potentiation, some exhibited depression, and others exhibited no change (Figure 3—example synapses; Figure A1—all synapses), with most exhibiting little change. These results show that a calcium-based plasticity rule determined from in vitro data can produce plasticity with spatially distributed in vivo-like synaptic input conditions.

### 3.2. Synaptic Plasticity Is Highly Robust to Trial-to-Trial Variability

During repeated behaviors, cortical neurons exhibit significant levels of trial-to-trial variability, yet it remains unclear how this variability affects synaptic plasticity. For synaptic plasticity to serve as the basis of learning, it should be robust to naturally observed variability in neuron spiking. However, many plasticity experiments use highly regular, precisely repeated stimulus patterns. To bridge the gap between in vitro plasticity findings using precisely repeated stimuli and in vivo plasticity with spatiotemporally dispersed inputs and trial-to-trial variability, we simulated the response to 10 repeated trials, with varying levels of trial-to-trial variability. This is analogous to 10 behavioral learning trials during which striatal neurons receive cortical input that varies from trial to trial.

For every level of trial-to-trial variability we simulated, a subset of synapses exhibited robust weight change at the end of 10 repeated trials. As shown in Figure 4A, synaptic weight over time consistently accumulates potentiation or depression for a subset of synapses regardless of variability level. This robust weight change also is observed when variability is produced by randomly moving spikes to different trains (Figure A2). These results predict that synaptic weight change is robust to high levels of trial-to-trial spike time variability, suggesting that in vivo variability does support synaptic plasticity.

Though synaptic weight change persisted across levels of trial-to-trial variability, the magnitude of weight change at the end of an experiment was reduced for increasing levels of variability. Figure 4B shows ending synaptic weight as a function of trial-to-trial variability, demonstrating that high variability reduces the magnitude of depression, but not potentiation. These results suggest that variability in spike timing primarily effects the magnitude of plasticity but rarely the direction (potentiation or depression).

### 3.3. Plasticity of a Single Synapse Is Only Partially Predicted by Its Pre-Synaptic Activity

Which properties of synaptic input patterns, that are potentially modulated by trial-to-trial variability, predict the magnitude and direction of synaptic plasticity? To investigate this question, we first asked whether the total pre-synaptic spike count for a given synapse predicted that synapse’s weight at the end of 10 trials (ending weight). We compared ending weight of every synapse to its total pre-synaptic spike count across all 10 trials for experiments at different levels of trial-to-trial variability (Figure 5A). We found a general pattern that synapses with low pre-synaptic spike counts exhibited little to no synaptic weight change; synapses with a moderate pre-synaptic spike count tended to exhibit depression; and synapses with a high pre-synaptic spike count exhibited potentiation. However, a wide range of ending synaptic weights was observed for synapses with moderate pre-synaptic spike counts, and further, these synapses were most affected by trial-to-trial variability. The synaptic weight was only weakly correlated with the synapse’s distance to the soma (Figure 5B). These results suggest that while spike count alone is a significant factor in predicting synaptic weight change, other spatiotemporal factors may also be important.

### 3.4. Plasticity of a Single Synapse Is Affected by Temporal Pattern of Pre-Synaptic Firing Rate

Though spike count per synapse was consistent for experiments, the trial-to-trial variability introduced changes in spike timing that altered the time-varying instantaneous pre-synaptic firing rate for each synapse. Thus, we next investigated the effect of the temporal pattern of pre-synaptic firing rate on synaptic plasticity. We computed a weight-change triggered average pre-synaptic firing rate by binning trials and synapses based on the magnitude of synaptic weight change following an individual trial, computing the instantaneous pre-synaptic firing rate vs. time for each synapse and trial, and averaging across synapse-trials within each bin.

Our results show that synapses that strongly potentiate exhibit a weight-change-triggered average pre-synaptic firing rate with a high peak firing rate late in the trial. In contrast, synapses that strongly depress exhibit an earlier peak firing rate or a moderate sustained pre-synaptic firing rate. Synapses with little or no weight change exhibit a low pre-synaptic firing rate (Figure 6A). This pattern was also observed when variability was introduced by moving spikes between trains (Figure A3A). We also calculated the weight-change-triggered average calcium concentration, to assess how pre-synaptic firing dynamics were translated into calcium elevations. As shown in Figure 6B and Figure A3B, synapses that potentiate have higher calcium concentration than synapses that depress, and synapses that strongly potentiate have the highest calcium. Though the largest differences are in the second half of the trial, weight change dependent differences in calcium concentration for synapses that depress are more apparent in the first half of the trial. These results suggest that input timing within a trial, which affects peak instantaneous firing rate and calcium concentration, is a critical factor for the direction and amplitude of synaptic weight change. Though temporal pattern discriminates strong LTP from strong LTD, the temporal patterns for moderate plasticity are not as clear. Thus, next we evaluated the role of spatial patterns of synaptic input.

### 3.5. Plasticity of a Single Synapse Is Affected by Neighboring Synaptic Activity

Prior work has shown that synaptic plasticity can be affected by spatiotemporally cooperative synaptic activity—that is, multiple synapses on the same dendritic branch active within a limited time window [20,23,78,79,80,81,82]. Our prior work has shown that spatiotemporal activity patterns have nonlinear, spatially specific effects on calcium transients in dendrites and spines [43]. Thus, it is likely that nearby synaptic activity can cooperatively influence plasticity in our calcium-based model. As shown in Figure 5, the number of input spikes to a given synapse is not sufficient to fully determine its synaptic weight change. Therefore, we next investigated the cooperative effect of neighboring synaptic activity on weight change at each synapse.

We repeated simulations using 125 different mappings of spike trains to clusters of synapses (two examples are show in Figure 2). Each simulation used a different number of spines per cluster and a different cluster length. Since the change in plasticity is consistent with repeated trials (Figure 4), we simulated a single trial. The weight-change-triggered pre-synaptic firing rate (Figure 7A) reveals that, similar to Figure 6, synapses that strongly potentiate have a transiently high firing rate toward the end of the trial. In addition, the calcium concentration (Figure 7B) exhibits a higher concentration later in the trial, as observed in Figure 6.

To examine the effect of spatial pattern, in Figure 8, we plot the weight-change-triggered average pre-synaptic firing rate of the 19 nearest neighbors for a randomly selected set of 25 mappings. Each panel shows the spatiotemporal pattern for a single weight-change bin. Figure 8 shows that synapses with larger weight change (either depression or potentiation) had high firing rates for a subset of neighbors. The smaller the change in weight, the lower the firing rate to neighboring synapses.

We evaluated several spatial properties that may influence change in synaptic weight, including cluster size, mean firing rate of neighbors and maximum firing rate of neighbors. To calculate mean firing rate of neighbors, for every synapse, its 19 nearest neighbors’ spike trains were combined and a single time series for instantaneous firing rate was calculated from the spike train representing all 19 nearest neighbors. Neither the mean firing rate of the neighbors (Figure 7C), nor the maximum firing rate across neighbors (not shown), exhibited a consistent pattern as a function of weight change. The correlation between weight change and maximum firing rate of neighbors or cluster length were both small, but significant (maximum firing rate: R = 0.046, *p* = 8.10 × 10^−12^; cluster length: R = −0.047, *p* = 1.49 × 10^−12^). However, the large number of observations and non-linear relationships may impede discovering spatiotemporal predictors of synaptic weight change. Thus, we used machine learning techniques to identify features that were predictive of synaptic weight change.

### 3.6. Pre-Synaptic Firing Rate and Cluster Length Predict Synaptic Plasticity

To quantitatively assess how spatiotemporal pattern of synaptic input controls synaptic plasticity, we used random forest regression to predict synaptic weight change using several features. First, we used the calcium dynamics to determine an upper bound on the regression score (a measure of predictive ability), since the learning rule directly depends on calcium. Then, we used instantaneous firing rate to the synapse, together with either firing rate of neighboring synapses, cluster length, spines per cluster or distance of synapse to the soma. Both the instantaneous firing rate and the calcium concentration were discretized into a small number of time samples to coarsely represent the mean over time: e.g., 1 sample measures the mean firing rate for the entire trial.

Figure 9A shows that calcium dynamics is an excellent predictor of the weight change. Using three time samples for the calcium dynamics improved the prediction of weight change compared to only one sample, but a further increase to five time samples was not helpful. The calcium at the end of the trial (i.e., the last sample) was more important, meaning it was the best predictor of the weight change (Figure 9B).

Figure 9C shows that the spatiotemporal pattern of synaptic input also can predict the synaptic weight change. Increasing the resolution of sampling the firing rate did not improved the prediction of weight change, despite the clear temporal pattern exhibited in Figure 6 and Figure 7. Adding in cluster length improved prediction regardless of the number of time samples of instantaneous firing rate. Prediction of weight change using mean firing rate and cluster length was almost as good as using calcium. None of the other spatial measures improved the prediction of ending synaptic weight. In summary, mean firing rate of direct input to the synapse is crucial for determining synaptic plasticity, and cluster length also influences direction and magnitude of synaptic plasticity.

To evaluate the role of within-trial correlations between neurons, we repeated simulations using spike trains in which interspike intervals (ISIs) were shuffled. Using the same number of synaptic inputs resulted in predominantly synaptic depression (mean synaptic weight change <1.0 for most simulations, Figure A4B). Thus, we increased the number of synaptic inputs with randomized ISIs to produce a similar mean weight change distribution as observed without randomizing ISIs (Figure A4C). The randomization changed the firing rate dynamics for synapses that potentiated, though not the pattern of calcium dynamics. Figure 10A shows a lack of high pre-synaptic firing rate at the end of the trial, though an increase in calcium over time is still apparent (Figure 10B).

Figure 10C,E show that either calcium dynamics, or firing rate dynamics and cluster length are good predictors of synaptic weight change, even in the absence of within-trial correlations between pre-synaptic inputs. As observed when correlations are maintained, an increase in the number of time samples of the calcium dynamics improves the prediction of weight change (Figure 10C), though in this case the middle samples have more importance than the last sample (Figure 10D). When using pre-synaptic firing rate as input, cluster length improved the prediction of ending synaptic weight, as observed using non-shuffled inputs. In summary, destroying between-neuron correlations by shuffling spike train ISIs greatly reduces the occurrence of synaptic potentatiation. However, if the number of synaptic inputs is increased, then potentiation is restored, along with the ability to predict the change in synaptic strength from either calcium concentration or pre-synaptic firing rate and cluster length.

### 3.7. Spike Timing Together with Pre-Synaptic Firing Rate Can Predict Instantaneous Weight Change

Numerous studies show that LTP versus LTD is determined from precise timing of pre- and post-synaptic spiking [7,12,19,30], though pre-synaptic firing rate also contributes when using in vivo-like spike trains [11,18]. Thus, the ability to predict weight change may improve if spike timing were used in addition to firing rate. During a 1 s trial, there may be numerous pairings of pre- and post-synaptic spikes; thus, we evaluated instantaneous weight changes as a function of interspike interval of the most proximal pre- and post-synaptic spikes preceding the weight change event as well as pre-synaptic firing rate for 50 ms prior to the weight change. Some research shows that multiple pre-post pairings can occur using in vivo spike trains [19], thus we also considered the pre-post interval using the second most proximal pre-synaptic spike, and the interval between the two pre-snaptic spikes (pre1-pre2).

Figure 11B shows that interspike interval influences instantaneous weight change in the manner observed in slice experiments. Decreases in weight have negative interspike intervals, increases in weight have positive interspike intervals, and the strongest potentation weight change bin has a mean interspike interval of 10 msec. We performed a random forest regression using interspike interval, combined with one or two additional features. Figure 11A shows that prediction of instantaneous weight change using interspike interval alone is not a good predictor of instantaneous weight change. This is due to the large range of weight changes at each ISI (Figure A5). Using ISI together with the pre-synaptic firing rate greatly improves the weight change prediction, more so than using the second ISI or pre1-pre2. Adding in pre1-pre2 as a third feature improved the prediction score still further, to a level comparable to predicting the ending weight from mean pre-synaptic firing. This shows that in the variable in vivo environment, both pre-synaptic firing rate and spike timing together determine the magnitude and direction of synaptic plasticity

## 4. Discussion

In this study, we addressed how temporal and spatial patterns of in vivo-like synaptic inputs with trial-to-trial variability impact the direction and magnitude of synaptic plasticity. We employed a calcium-based plasticity rule in a biologically constrained, morphologically complete SPN model, previously validated on in vitro plasticity protocols, to predict plasticity for in vivo-like conditions. We found that synaptic plasticity with in vivo-like activity is robust to trial-to-trial variability. Further, we found that both pre-synaptic firing rate and the size of clusters of synaptic inputs on the dendritic tree controls synaptic plasticity outcomes. Calcium dynamics, on a slower time scale than used for the learning rule, was highly predictive of the change in synaptic weight. Calcium dynamics are influenced by local synaptic activity and depolarization. Accordingly, neighboring activity appears to influence the magnitude of synaptic plasticity, in that synapses with large weight changes have a few neighbors that receive a high rate of synaptic inputs. However, we were unable to find a quantitative measure of neighboring activity that improved weight change prediction by the random forest regression. Our work provides key insights into the nature of synaptic plasticity in conditions more likely to occur during behavior.

Our finding that plasticity at a single synapse is influenced by neighboring synaptic activity and size of clusters is consistent with prior studies (both in vivo and in vitro) showing relationships between spatial synaptic input patterns and plasticity. For instance, in SPNs in vitro, spatially clustered synaptic inputs can produce supralinear depolarization and calcium transients [39,43,71,76,83]. In cortical and hippocampal pyramidal neurons in vitro, spatially clustered inputs have been shown to induce synaptic potentiation (termed cooperative LTP) [20,23,24,81,82,84,85,86,87,88]. Complementing these in vitro findings, correlated activity in spatially clustered spines has been observed in vivo in pyramidal neurons [9,89,90,91], and in vivo calcium transients in neighboring spines of the same dendritic branch correlate with structural potentiation of spines in motor cortical neurons [80]. In the striatum, a recent study showed that SPN spines that are innervated by motor cortex pyramidal neurons involved in learning are clustered on dendrites [92]. Building on these findings, our work predicts that neighboring synaptic interactions can influence the direction and magnitude of corticostriatal synaptic plasticity in vivo, with high neighboring activity producing a larger magnitude of both LTP and LTD, and smaller clusters tending to produce LTP. Our results further suggest that correlated activity between spike trains allows synaptic plasticity to occur with fewer synaptic inputs. When the correlations were eliminated by shuffling ISIs, LTP did not occur, but LTP was recovered by increasing the number of spike trains. In other words, coordination of firing between pre-synaptic neurons reduces the number of pre-synaptic spike trains needed to produce LTP. The mean distance between spines in the experimentally observed clusters on SPN dendrites was ~10 μm [92], which is consistent with our observation that smaller clusters are more likely to exhibit LTP.

Functional synaptic clustering is a key component in plasticity and learning; however, it is unclear whether functional clustering arises from synaptic connectivity in development, or if activity dependent plasticity can generate functional clusters starting from random connectivity. By using randomly distributed inputs, our results suggest that functional synaptic clusters may emerge from potentiating neighboring synapses that each receive higher than average input. Our results are consistent with another modeling study investigating the impact of synaptic clustering on somatic membrane potential with in vivo-like conditions [93], which suggested that global plasticity rules would *not* be sufficient for formation of synaptic clusters. Our data-driven model of calcium dynamics accounts for global processes (produced by somatic spiking) and local processes (such as NMDA spikes), and the calcium-based synaptic plasticity rule can produce spatial clustering.

Our work predicts that synapse specific potentiation will occur for spatially clustered synapses during the induction phase of plasticity. However, a second critical component for synapse-specificity involves the maintenance phase of long term synaptic plasticity. Recent theoretical work [94] has addressed the maintenance phase, predicting that spatially clustered synapses will maintain specificity if a molecular switch for synthesis of synaptic proteins is limited to the spine head. Their work highlights the importance of plasticity models that account for dendritic spine mechanisms, consistent with our research.

A common concern about data-driven computational models is that they are underconstrained, and thus it is important to show that model results are robust to parameter variation. Our model of a direct pathway striatal SPN is directly based on our prior one [76] which did vary parameters, and, importantly, showed qualitatively consistent synaptic integration (duration of upstate and firing rate in response to synaptic inputs) for a large number of parameter variations. Furthermore, by holding the model constant while varying the inputs, we can better understand the effect of input variations. Nonetheless, future studies should evaluate synaptic plasticity in other direct pathway SPN model variants, in order to better compare with development of synaptic plasticity in striatal SPNs of the indirect pathway.

Our results suggest a mechanism for spatially balanced potentiation and depression, which could provide homeostatic balance and prevent runaway potentiation. Synapses with small increases in calcium, produced by nearby depolarization and/or weak synaptic inputs, exhibited synaptic depression. These observations are consistent with in vivo experiments that reveal spine shrinkage in inactive spines accompanying structural potentiation of nearby spines, suggesting heterosynaptic depression as an important compensatory plasticity mechanism [95,96]. Our calcium-based plasticity rule effectively captures the impact of neighboring synaptic activity on depression to support compensatory plasticity. Predictions from our results could be tested experimentally in vitro using glutamate uncaging to apply low frequency inputs to neighboring dendritic spines, together with calcium imaging of multiple dendritic spines to relate stimulation patterns to calcium activity and synaptic plasticity. Alternatively, development of methods for optically stimulating synapses with specified spatiotemporal patterns may be able to test our model predictions in the future.

This study has important implications for plasticity in variable in vivo conditions by showing that precise spike timing is not required. It is clear from numerous in vivo studies that neither the frequency based protocols nor the spike timing dependent protocols used in brain slices exactly replicate the patterns of synaptic inputs that occur during behavioral learning. Instead, these protocols are tools that are useful for understanding the molecular mechanisms underlying synaptic plasticity.

Similar to a non-spatial model of pyramidal neurons [18], our work suggests that while plasticity is sensitive to firing rate, it is highly robust to variance in precise spike timing during a trial. This robustness to spike timing may follow from the sensitivity to spike rate when using in vivo-like spike trains, as observed in pyramidal neurons [11,18]. In addition, robustness to spike timing may result from a large number of pairings of pre-synaptic and post-synaptic stimuli, as shown in an in vitro study that used jittered spike timings [97]. Together, these modeling and experimental studies extend the usefulness of in vitro synaptic plasticity studies. In summary, given the variability of spike timing [98], in vivo plasticity likely depends on both precise spike timing rules as well as firing frequency and neighboring activity.

Our research is a major advance over prior work which implemented a plasticity rule based on simplified calcium dynamics in a non-spatial model [33] or derived firing rate plasticity models from simplified calcium dynamics [99]. First, our calcium-based plasticity rule is not explicitly dependent on spike timing, though we previously showed it could reproduce in vitro spike timing dependent plasticity results. Second, our calcium-based rule is coupled to a data-driven biophysical model of calcium dynamics in a neuron model that includes dendritic morphology. By using a spatial model, we include two of the mechanisms previously recommended to unify synaptic plasticity rules [77]: temporal properties of calcium signaling and locally spreading (dendritic) signals. By implementing biophysical mechanisms controlling calcium dynamics, we avoid making assumptions about the source of the calcium and minimize the phenomenological aspects of our model, which are limited to the rule itself. Nonetheless, our work is consistent with prior models that predicted that plasticity was not sensitive to spike timing with in vivo-like firing patterns [18]. More importantly, we extend prior work to show that temporal patterns are still important, and that spatial interactions among synapses influence plasticity.

The distinct temporal patterns that were associated with potentiation or depression have implications for striatal and wider basal ganglia circuit function. Though corticostriatal LTP requires dopamine in addition to calcium elevation [7], prior experimental work has shown calcium-dependent synaptic eligibility traces in SPNs [100]. These eligibility traces for corticostriatal LTP exhibit a temporal dependence such that an LTP protocol followed within a few seconds by dopaminergic stimulation produces LTP [25]. Thus, though our plasticity model does not account for dopamine directly, we suggest that our calcium-based plasticity rule accounts for eligibility traces for LTD or LTP and captures the spatiotemporal pattern of corticostriatal activity that, when followed by dopaminergic stimulation, produces plasticity. Further, as dopaminergic activity is consistent with volume transmission [101,102], we suggest that our calcium-based plasticity rule provides the spatiotemporal specificity indicating which synapses are eligible for reinforcement when followed by a spatially diffuse dopamine signal.

Another consequence of our research is to provide a mechanism for the observed decreasing trial-to-trial variability in cortical and striatal neurons in vivo during learning [103]. Corticostriatal plasticity may occur even with highly variable conditions early in learning and then, by potentiating the relevant synapses, produce decreased variability in striatal spiking with learning.

Our results show that the spatial aspects of synaptic integration may contribute to synaptic plasticity. However, spiking network models that incorporate spike-timing dependent plasticity rules to investigate the effect of plasticity on network activity neglect spatial patterns of synaptic inputs to a single neuron, reducing neurons to point processes [104,105,106]. We suggest that our calcium-based plasticity rule could be used in future work to develop simplified plasticity models incorporating both temporal and spatial effects of synaptic activity. A statistical model using temporal and spatial kernels to predict the effect of both direct and neighboring synaptic activity on the synaptic weight of each synapse could be derived from biophysical single neuron models with our calcium-based rule and then simulated efficiently in large networks with simplified neuron models. Incorporating this synaptic plasticity rule in large scale simulations of network models of the basal ganglia could then better predict how corticostriatal plasticity supports goal-directed and habit learning and identify potential therapeutic targets for modulating aberrant plasticity in addiction.

## Figures and Tables

**Figure 1 biomolecules-12-01402-f001:**
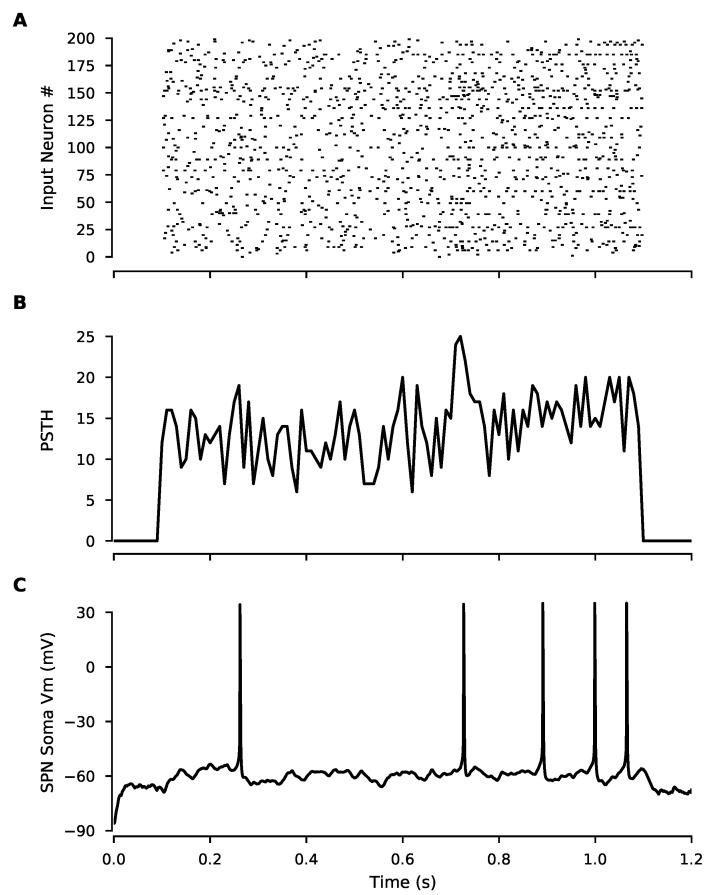
In vivo-like inputs constructed from cortical spike trains produce spiking in SPNs. (**A**) Raster plot shows spike times for each cortical input in the model, constructed from in vivo spike train recordings. (**B**) Peri-stimulus time histogram of the above raster plot (spike counts per 10 ms bin) (**C**) Somatic membrane potential of the SPN model showing spiking output induced by cortical input.

**Figure 2 biomolecules-12-01402-f002:**
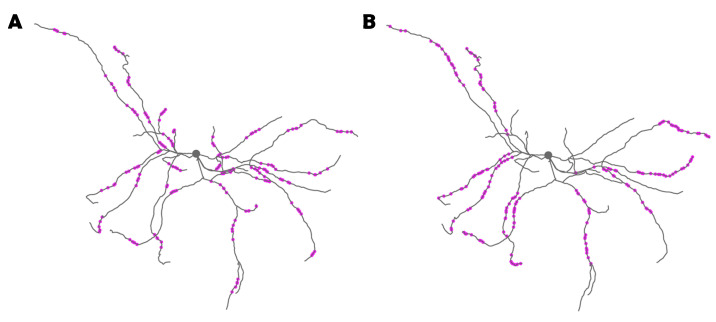
Synapses receiving spike trains with clustered input. (**A**) Spike trains target dense clusters of 5 synapses. (**B**) Spike trains target dispersed clusters of 40 synapses.

**Figure 3 biomolecules-12-01402-f003:**
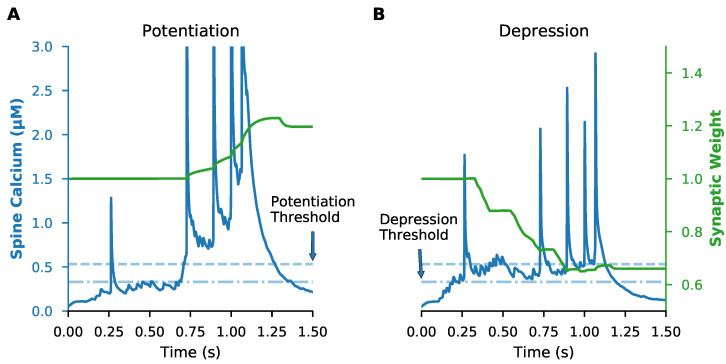
Calcium-based plasticity rule produces both potentiation and depression. A calcium-based plasticity rule was implemented with dual amplitude and duration thresholds. LTD required that spine calcium concentration exceed the amplitude threshold (dot-dashed line) of 0.33 μMolar for greater than 28 ms, while LTP required that spine calcium concentration exceed a higher amplitude threshold (dashed line) of 0.53 μMolar for at least 3.3 ms. Example traces are shown of a synapse that potentiates (**A**) or depresses (**B**) following a single trial. Blue lines show spine calcium concentration (with left y-axis) and green lines show synaptic weight (with right y-axis) for each example synapse.

**Figure 4 biomolecules-12-01402-f004:**
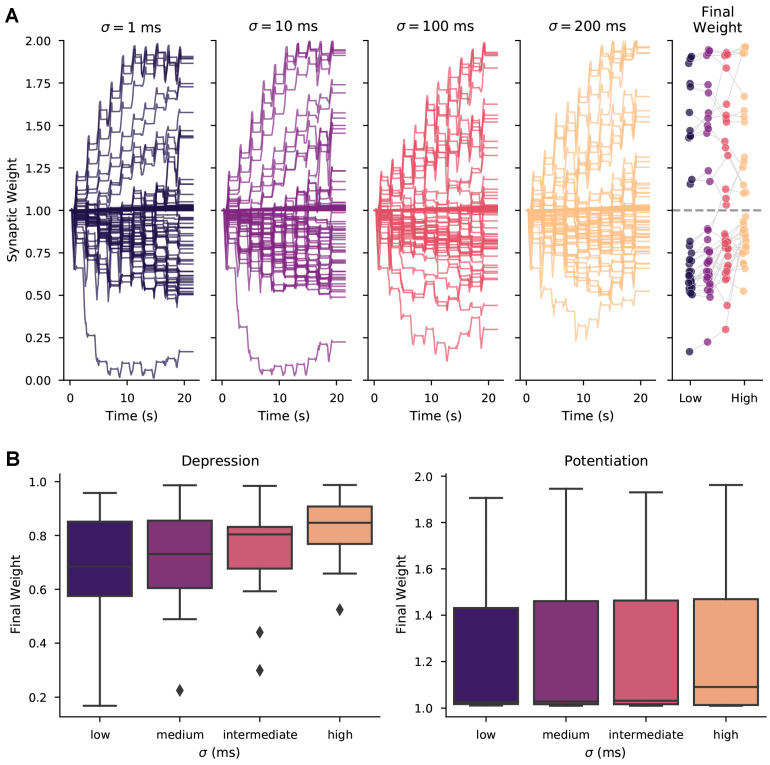
Synaptic plasticity is robust to trial-to-trial variability. (**A**) The weight of every synapse over a full 10-trial experiment is shown for different trial-to-trial variability conditions, with the σ value corresponding to the standard deviation of random jitter of spike times. (Right) Ending weight of each synapse is shown for each variability condition (synapses with near zero weight change excluded for visualization). (**B**) Distribution of final weights grouped by potentiation and depression shows that variability reduces synaptic depression magnitudes, but has little effect on the distribution of potentiation magnitudes. Correlation of ending synaptic weight versus variability was significant for depressing synapses (R = 0.306, *p* = 0.0006, N = 121 events), but not for potentiating synapses (R = 0.054, *p* = 0.510, N = 148 events).

**Figure 5 biomolecules-12-01402-f005:**
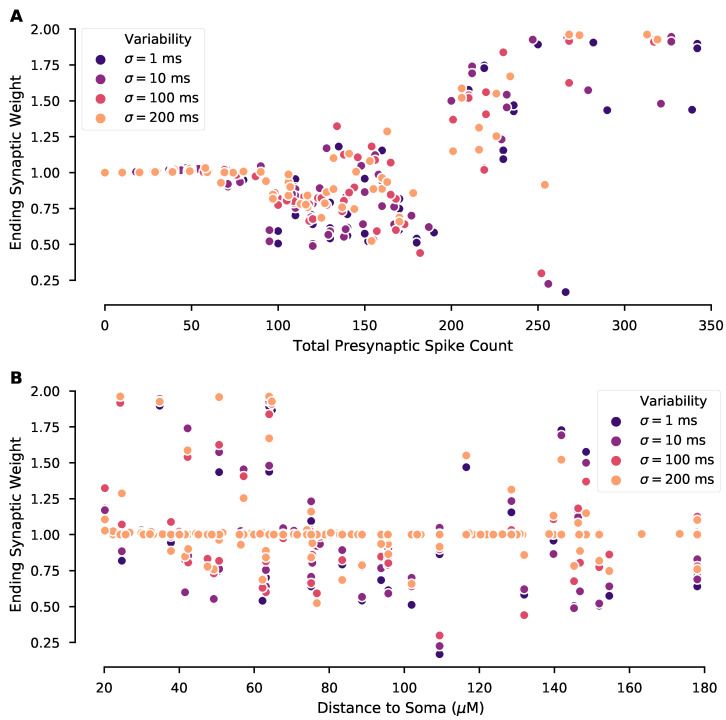
Ending synaptic weight is partially predicted by total pre-synaptic spike count per synapse. Ending synaptic weight of each synapse is plotted versus (**A**) its total pre-synaptic spike count across all 10 repeated trials and (**B**) distance of the synapse from the soma, with experiments separated by the level of trial-to-trial variability. (**A**) Ending weight exhibits no change for low spike counts, tends toward depression for intermediate spike counts, and exhibits potentiation for high spike counts. This trend is consistent regardless of trial-to-trial variability; however, for intermediate spike counts the ending weight is highly variable. (**B**) Correlation between ending synaptic weight and distance to soma is not significant for σ = 100 ms (R = −0.23, *p* = 0.07) or 200 ms (R = −0.21, *p* = 0.11), but is significant for σ = 1 ms (R = −0.25, *p* = 0.04) and 10 ms (R = −0.25, *p* = 0.04).

**Figure 6 biomolecules-12-01402-f006:**
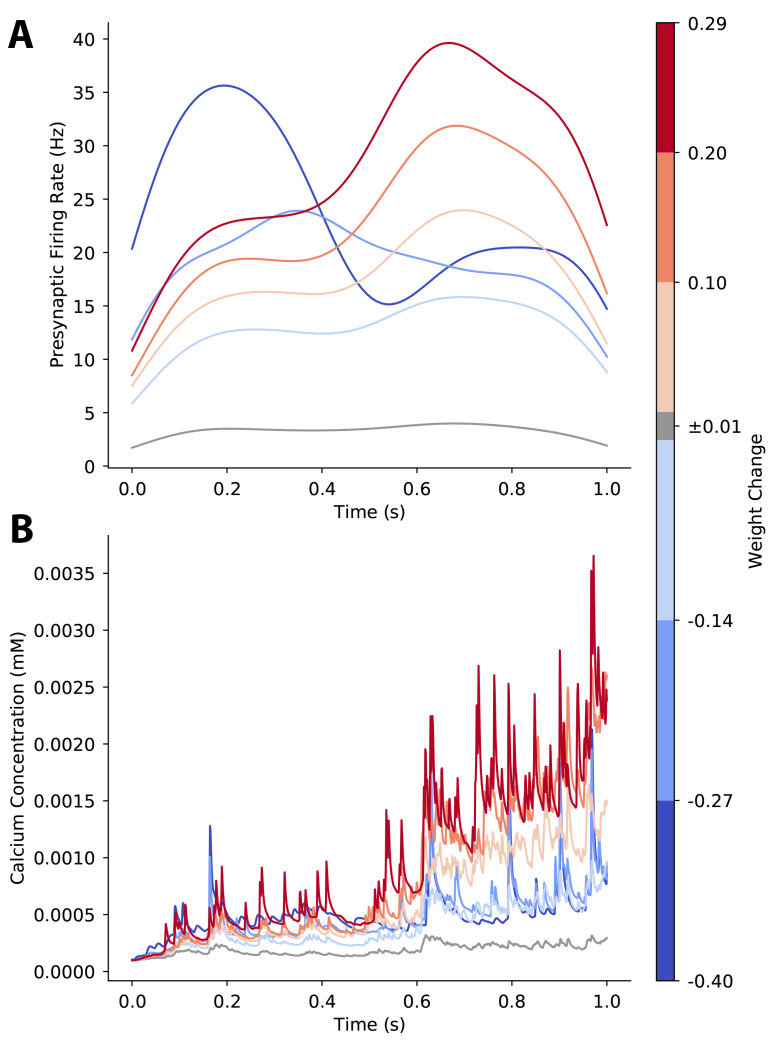
Temporal pattern determines direction of plasticity as shown by weight-change-triggered-average. (**A**) For each synapse on each trial, we computed the instantaneous firing rate of its pre-synaptic input activity and grouped synapse-trials into bins based on the size of the synaptic weight change that occurred following a single trial. Then, we averaged across the instantaneous firing rate of each bin. Late and high peak firing rates lead to LTP, while earlier peak firing rate or moderate firing rate leads to LTD. (**B**) For each synapse on each trial, we computed the calcium concentration for each synapse-trial and averaged across the calcium concentration for each weight change bin. Calcium concentration is higher during the second part of the trial, both for potentiating and depressing synapses.

**Figure 7 biomolecules-12-01402-f007:**
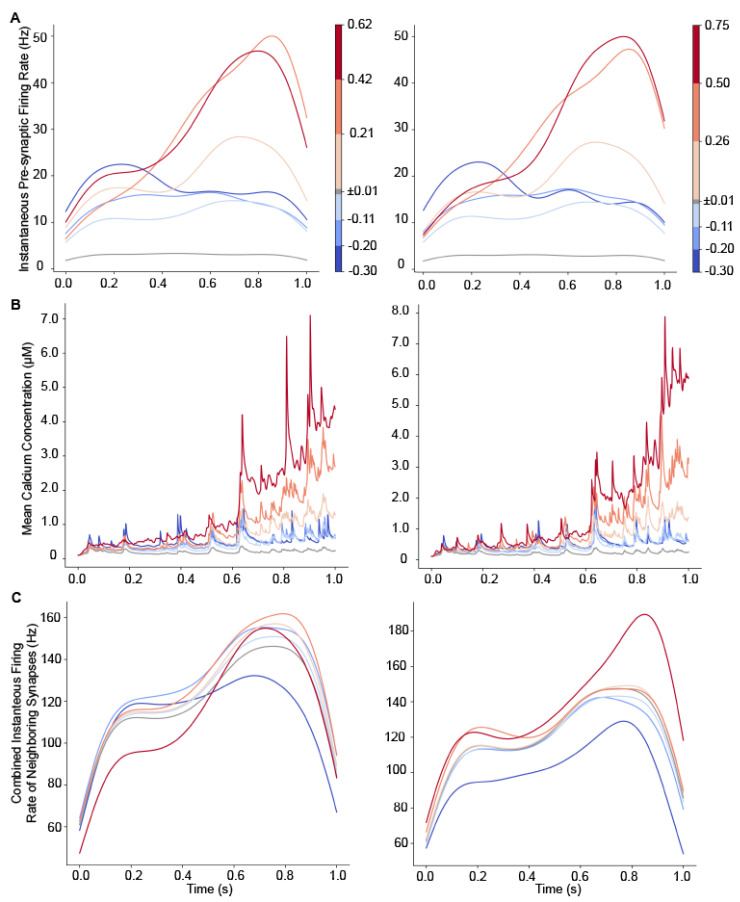
Temporal pattern of input predicts weight change for different variations of the mapping from spike trains to synapses. Weight change triggered average pre-synaptic firing rate and calcium concentration for two different sets of mappings from spike trains to synapses. (**A**) Weight change triggered pre-synaptic firing rate for two different sets of 24 different mappings. Synapses that potentiate have a transiently high firing rate at the end of the trial. (**B**) Calcium concentration for two different sets of 24 different mappings. Calcium concentration determines direction of plasticity as shown by the weight-change-triggered-average. Regardless of whether peak synaptic firing rate occurs early or late in the trial, calcium concentration is highest during the second half of the trials. (**C**) Weight change triggered pre-synaptic firing rate of 19 neighboring synapses. For synapses in each weight change bin, the spike trains for the 19 neighboring synapses are combined into a single train, and then instantaneous firing rate is calculated from that single train.

**Figure 8 biomolecules-12-01402-f008:**
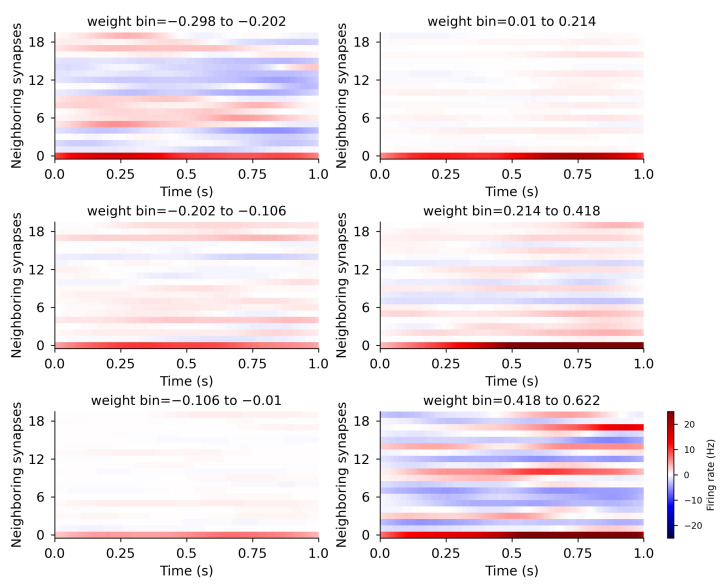
Activity of neighboring synapses influences magnitude of plasticity. spatiotemporal firing patterns produce different effects on the magnitude of synaptic plasticity. Each panel shows the mean subtracted instantaneous firing rate following a trial for a different weight change bin. Neighboring synapses 1–19 (y-axis) shows firing rate (color coded) versus time (x-axis) of 19 nearest neighbors to synapse 0. Synapses with small changes in weight have neighbors that receive average input firing rates. Synapses with intermediate to high weight changes have a few neighbors with higher than average input firing rates.

**Figure 9 biomolecules-12-01402-f009:**
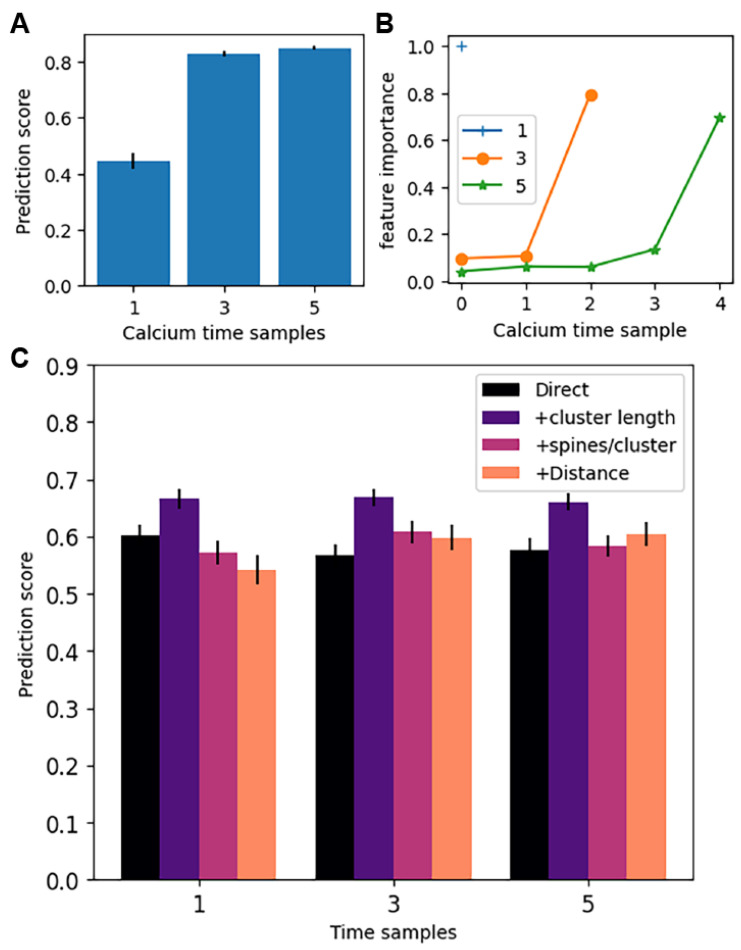
Spatiotemporal characteristics predict weight change. Prediction score is the coefficient of determination, R${}^{2}$, of the predicted weight change for the test set. For each of 5 sets of 25 simulations, we used 75% of the data as the training set and the remainder as test set, repeated 4 times giving a total of 20 regressions for each combination of features. Error bars show 1 standard error. (**A**) ANOVA shows that increasing number of time samples of calcium concentration improves the prediction score (F(2,60) = 165.7, *p* = 1.78 × 10^−24^). Post-hoc *t*-test shows that 3 samples is significantly better than 1 sample (*p* = 2.84 × 10^−15^). (**B**) Feature importance shows that the last calcium time sample is the most important for predicting weight change. Note that the feature importances must sum to 1, so the feature importance for one calcium time sample is 1.0 by default. (**C**) A second set of random forest regressions used time samples of pre-synaptic firing frequency (direct) plus one other feature. An ANOVA on the other feature (none, cluster length, spines per cluster, and distance to soma) was significant (F(3,236) = 12.89, *p* = 7.88 × 10^−8^). Post-hoc *t*-test shows that cluster length, but no other feature, significantly improved the prediction score (*p* = 2.2 × 10^−7^).

**Figure 10 biomolecules-12-01402-f010:**
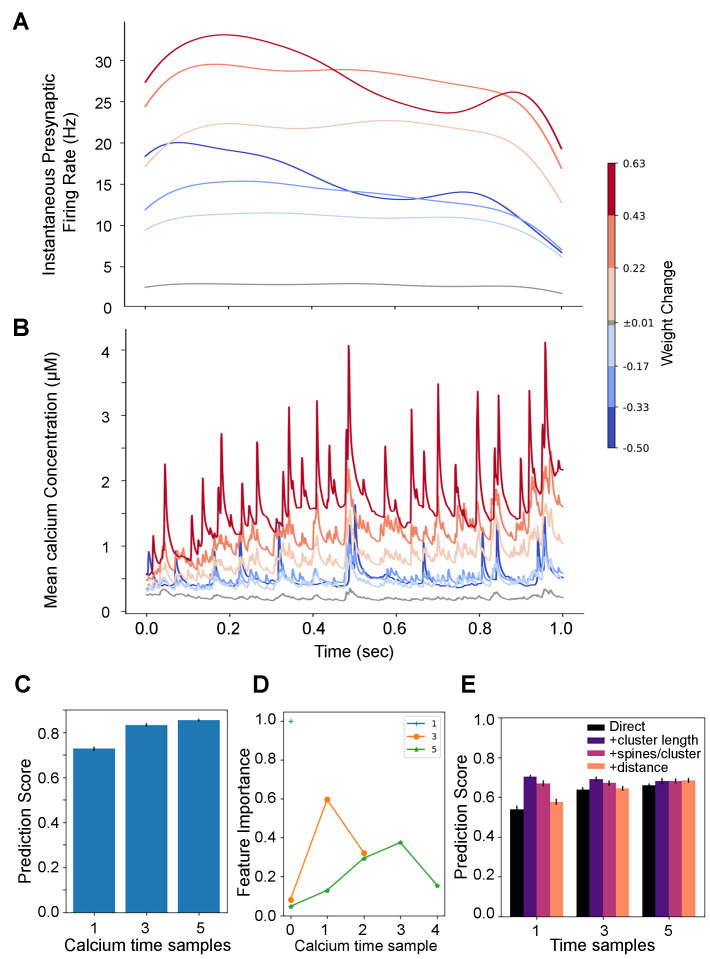
Simulations using spike trains with shuffled ISIs. Temporal pattern of pre-synaptic firing rate and calcium dynamics are different with shuffled spike trains, but still predict weight change. (**A**,**B**) The end weight of one set of 25 simulations using 208 spike trains with shuffled ISIs was subdivided into seven weight change bins. (**A**) Weight change triggered average pre-synaptic firing rate distinguishes weight change bins but differs from the temporal pattern of non-shuffled spike trains. (**B**) Weight change triggered average calcium concentration exhibits temporal pattern similar to non-shuffled spike trains, with higher values later in the trial for synapses that potentiate. (**C**–**E**) For each of 5 sets of 25 simulations, we used 75% of the data as the training set and the remainder as test set, repeated 4 times, giving a total of 20 regressions for each combination of features. (**C**) Random Forest regression using calcium time samples as inputs is better with 3 or 5 time samples compared to 1 time sample. (**D**) Feature importance from random forest regression using calcium time samples as inputs shows that the middle samples are more important than the ending sample. (**E**) Random Forest regression using time samples of pre-synaptic firing and spatial feature as inputs shows that using 3 time samples has better prediction score than 1 time sample. Cluster length improves the prediction score independent of the number of time samples.

**Figure 11 biomolecules-12-01402-f011:**
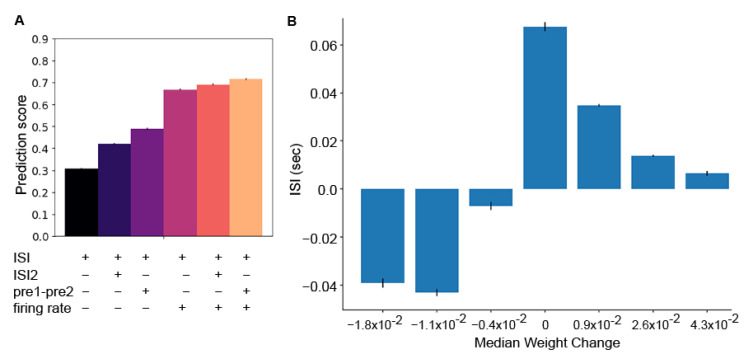
Both ISI and pre-synaptic firing rate are required to predict instantaneous synaptic weight change. Instantaneous weight change events were grouped into 7 weight change bins. Interspike interval was calculated from the closest pre-synaptic spike and closest post-synaptic spike that preceded the weight change event. (**A**) ISI and pre-synaptic firing rate together yielded the best two-variable prediction score. Adding in the interval between the two pre-synaptic spikes (pre1-pre2) produced a prediction score comparable to, but not better than the end weight prediction score. Note that a subset of instantaneous weight changes were not preceded by two pre-synaptic spikes and thus were excluded from those random forest regressions. (**B**) The mean ISI was calculated for each weight change bin. Note that the highest weight increase had small positive interspike intervals, and smaller weight increases had larger postive interspike intervals. As predicted by STDP theory, the weight decreases had negative interspike intervals.

## Data Availability

All code for simulation and analysis is available on github https://github.com/neurord/moose_nerp/releases/tag/v2.1 (accessed on 6 June 2022) and modelDB (accession number 267552).

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
