# Peer review of "Synaptic Plasticity Is Predicted by Spatiotemporal Firing Rate Patterns and Robust to In Vivo-like Variability"

_biomolecules, 2022, doi:10.3390/biom12101402_

Round 1
Reviewer 1 Report
This is a well thought out computational study that illustrates the robustness of their postulated rules for synaptic LTP and LTD to simulated in vivo variability in spike train inputs. They elaborate on the need for clustered input to produce reliable changes on synaptic weight, and the homeostatic effect of LTD counteracting LTP in neighboring clusters. I have only a few minor suggestions.
1) State why anterior lateral motor cortical spike trains were chosen. At a behavioral level, what is plasticity at the cortico-striatal synapse encoding? Learning a sequence of actions to obtain reward? Or simply habit learning, since dopamine is not mentioned until line 523? I think the big picture implications should appear in the introduction. Is the Ca2+ eligibility trace being modeled separately from the dopamine (and possibly ACh pauses required for plasticity (see Fig. 4 of (Reynolds et al., 2022)
Reynolds JNJ, Avvisati R, Dodson PD, Fisher SD, Oswald MJ, Wickens JR, Zhang Y-F (2022) Coincidence of cholinergic pauses, dopaminergic activation and depolarisation of spiny projection neurons drives synaptic plasticity in the striatum. Nat Commun 13:1296.
A mention of how the present study related to Reynolds et al 2022 above would be helpful to put the work in context.
2) Briefly restate the premise of the synaptic plasticity rule in the introduction: large, short elevation in Ca2+ produced by spatially and temporally coherent input produces LTP, whereas longer lasting but lower elevation of Ca2+ in areas surrounding the hot spots produces LTD to keep the overall summed synaptic weights in a homeostatic state.
3) The level of biophysical detail and model calibration is impressive. However, I presume the model parameters are nonetheless underconstrained. Therefore rather than a single exemplar gold standard SPN model, shouldn’t several examplars be generated? Perhaps not for this study, but it might be worth a discussion point.
line 54, put the abbreviation (SP) after spiny projection neuron.
I do not see 19 traces (line 344) nor 20 (line 349) in each panel of figure 7
Author Response
This is a well thought out computational study that illustrates the robustness of their postulated rules for synaptic LTP and LTD to simulated in vivo variability in spike train inputs. They elaborate on the need for clustered input to produce reliable changes on synaptic weight, and the homeostatic effect of LTD counteracting LTP in neighboring clusters. I have only a few minor suggestions.
1) State why anterior lateral motor cortical spike trains were chosen.
We selected anterior lateral motor cortical spike trains because (1) this brain region projects to the striatum and (2) when the work was initiated, this was the only dataset on crcns.org from a brain region that projects to the striatum. We now provide the first rationale in the methods, lines 86-88, and provide two additional references showing the importance of ALMC to striatal processing.
At a behavioral level, what is plasticity at the cortico-striatal synapse encoding? Learning a sequence of actions to obtain reward? Or simply habit learning, since dopamine is not mentioned until line 523? I think the big picture implications should appear in the introduction. Is the Ca2+ eligibility trace being modeled separately from the dopamine (and possibly ACh pauses required for plasticity (see Fig. 4 of (Reynolds et al., 2022).
A mention of how the present study related to Reynolds et al 2022 above would be helpful to put the work in context.
The reviewer is correct in that our synaptic plasticity represents the calcium eligibility trace, which is involved both in reward learning and habit learning. We now discuss how our research relates to Reynolds et al. 2022 in introduction, lines 47-56.
2) Briefly restate the premise of the synaptic plasticity rule in the introduction: large, short elevation in Ca2+ produced by spatially and temporally coherent input produces LTP, whereas longer lasting but lower elevation of Ca2+ in areas surrounding the hot spots produces LTD to keep the overall summed synaptic weights in a homeostatic state.
We now introduce the synaptic plasticity rule in the introduction (lines 70-72) “According to this rule, if the calcium exceeds the LTP threshold for the LTP duration criterion, the synaptic strength increases. If the calcium remains between the LTD and LTP thresholds for the LTD duration criterion, then synaptic strength decreases.” and explain the relationship to underlying signaling pathways, lines 61-64. “Both models and experiments suggest that a prolonged, but moderate elevation produces LTD by activation of the protein phosphatases or production of endocannabinoisds, and a shorter but higher amplitude calcium produces LTP through activation of protein kinases.”
3) The level of biophysical detail and model calibration is impressive. However, I presume the model parameters are nonetheless underconstrained. Therefore rather than a single exemplar gold standard SPN model, shouldn’t several examplars be generated? Perhaps not for this study, but it might be worth a discussion point.
This is an excellent point. We are well aware of issues with parameter optimization and underconstrained parameters. This model is directly based on our prior one (Prager, Dorman et al. Neuron 2020) which did vary parameters, and, importantly, showed qualitatively consistent synaptic integration (duration of upstate and firing rate in response to synaptic inputs) for a large number of parameter variations. By holding the model constant in this paper while varying the inputs, we can better understand the effect of input variability. We did not use additional exemplars due to computational resources, since the number of input variations and duration of trials already requires long simulation times on the NSG. Additional evidence of robustness of our results is provided by another publication (Jedrzejewska-Szmek, et al. Eur J Neurosci 2017) which used a slightly different model to show that different spike timing dependent plasticity protocols can produce synaptic plasticity using this calcium based plasticity rule. Using multiple examplars would be critical for comparing plasticity rules between direct and indirect pathway SPNs. We have added some of these points to the discussion, lines 503-511.
line 54, put the abbreviation (SP) after spiny projection neuron.
We added the abbreviation, SPN to the first place we use spiny projection neuron (Line 47).
I do not see 19 traces (line 344) nor 20 (line 349) in each panel of figure 7
Line 344: The 19 nearest neighbors are shown in Figure 8, not figure 7. We have changed the preceding lines 359-360 to clarify “In figure 8, we plot the weight-change-triggered …”
We also add (line 361) that” Each panel shows the spatio-temporal pattern for a single weight-change bin”. For each panel, the x axis is time, the y axis is the synapse number and the color represents the firing rate. This has been clarified in the caption to figure 8.
Line 349: we now state, “a single time series for instantaneous firing rate was calculated from the spike train representing all 19 nearest neighbors” (lines 367-369), and we also clarify by adding a sentence to the caption of figure 7C.
Reviewer 2 Report
An advantage of computer modeling is to assist in the leap from in vitro measures to the in vivo reality. This paper represented an important effort to do just that in the case of timing-related potentiation, including STDP, by evaluating responses to non-stationary in vivo motor cortex spike trains in a Ca-based plasticity model derived from slice STDP experiments.
My biggest concern with the paper is that the message remains obscure. It appears that the research was unable to demonstrate strict timing-based plasticity in the in vivo context. There may be several reasons for this. Perhaps the in vitro results are artifactual and not relevant to functional neural plasticity? Alternatively, plasticity mechanisms in striatum might differ from those of hippocampus and other tissues where timing-based has been assessed. Most likely, the current exploration is incomplete and their remains more to be done in order to demonstrate relevance? -- negative results are fine but should be labeled as such. Also, in vivo derived spike trains are not the trains that would be received by a particular striatal cell so that there is no relation between the various inputs. Therefore, rather than jitter inputs, it might be interesting to gradually alter the degree of correlations between inputs.
"Classical" STDP protocols demonstrated that a presyn-activation before postsyn-spike inside a time window gave LTP, reverse giving LTD. STDP is first mentioned in the results but the authors show no evidence that pre vs post spike times play a role in the plasticity shown. My understanding is that most of the spike-timing plasticity literature is based on STDP and not on coincident activation at neighboring synapses -- either way this should be addressed in the introduction. Do we consider STDP passe since Shouval etal 2010 paper? Else, it would seem valuable to explicitly test it in this study.
I couldn't find a description of the dendritic channel distribution, or the synaptic GABAA distribution. Channel distribution is only weakly fitted by depending on somatic recording so one would expect considerable uncertainty (as well as degeneracy), leading to multiple good-enough models. One would expect that some of these models would show better time-based effects than others. One could even optimize to produce good time-based results -- such a result would make predictions, partly testable, regarding channel distribution. Also please share ModelDB (with passwd) or github for the model to evaluate for reproducibility.
"Cortical spike trains indeed produced synaptic plasticity" -- to what extent is this a stochastic consequence of the time course of the mechanism, frequency and poisson lambda of the inputs and frequency and pattern of postsynaptic firing, etc? Can it be shown that there is more or different plasticity than would be predicted? This also relates to above question about train cross-correlations.
minor:
Fig 1 -- please show locations of these trains on the dendritic tree for both the more clustered and less clustered cases.
Having too many figures dilutes the message. Suggest moving more figures into the appendix.
Please share your "extended version of a parameter optimization algorithm we developed" (unless being prepared for a separate pub?)
Related recent paper on effect of neighboring syns -- Huertas et al (2022). Conditions for Synaptic Specificity during the Maintenance Phase of Synaptic Plasticity. eNeuro.
"full spine density observed experimentally" -> "full spine density observed anatomically"
What was the input into GABAA? Were there AMPA synapses?
Round 2
Reviewer 2 Report
improved